# Reliability of IMU-Based Gait Assessment in Clinical Stroke Rehabilitation

**DOI:** 10.3390/s22030908

**Published:** 2022-01-25

**Authors:** Richard A. W. Felius, Marieke Geerars, Sjoerd M. Bruijn, Jaap H. van Dieën, Natasja C. Wouda, Michiel Punt

**Affiliations:** 1Research Group Lifestyle and Health, Utrecht University of Applied Sciences, 3584 CS Utrecht, The Netherlands; Mgeerars@axioncontinu.nl (M.G.); N.Wouda@dehoogstraat.nl (N.C.W.); Michiel.Punt@hu.nl (M.P.); 2Faculty of Human Movement Sciences, Vrije Universiteit Amsterdam, 1081 HV Amsterdam, The Netherlands; s.m.bruijn@gmail.com (S.M.B.); j.van.dieen@vu.nl (J.H.v.D.); 3Physiotherapy Department Neurology, Rehabilitation Center de Parkgraaf, 3526 KJ Utrecht, The Netherlands; 4Physiotherapy Department Neurology, De Hoogstraat Revalidatie, 3583 TM Utrecht, The Netherlands

**Keywords:** cerebral vascular accident, sensors, walking, recovery, accelerometer, gait quality, neurological disorder, functional gait assessment

## Abstract

Background: Gait is often impaired in people after stroke, restricting personal independence and affecting quality of life. During stroke rehabilitation, walking capacity is conventionally assessed by measuring walking distance and speed. Gait features, such as asymmetry and variability, are not routinely determined, but may provide more specific insights into the patient’s walking capacity. Inertial measurement units offer a feasible and promising tool to determine these gait features. Objective: We examined the test–retest reliability of inertial measurement units-based gait features measured in a two-minute walking assessment in people after stroke and while in clinical rehabilitation. Method: Thirty-one people after stroke performed two assessments with a test–retest interval of 24 h. Each assessment consisted of a two-minute walking test on a 14-m walking path. Participants were equipped with three inertial measurement units, placed at both feet and at the low back. In total, 166 gait features were calculated for each assessment, consisting of spatio-temporal (56), frequency (26), complexity (63), and asymmetry (14) features. The reliability was determined using the intraclass correlation coefficient. Additionally, the minimal detectable change and the relative minimal detectable change were computed. Results: Overall, 107 gait features had good–excellent reliability, consisting of 50 spatio-temporal, 8 frequency, 36 complexity, and 13 symmetry features. The relative minimal detectable change of these features ranged between 0.5 and 1.5 standard deviations. Conclusion: Gait can reliably be assessed in people after stroke in clinical stroke rehabilitation using three inertial measurement units.

## 1. Introduction

Walking dysfunction is a common problem in people after stroke, restricting personal independence and affecting quality of life [1,2]. Walking dysfunction in people after stroke is characterised by decreased walking speed, shorter stride length, and gait asymmetry [3,4]. These changes in gait patterns are related to a higher fall risk among the elderly and people after stroke [5,6,7,8,9]. Falling can result in physical injury, emotional dysfunction, and is the number one cause of unexpected death [10]. Thus, to promote quality of life and reduce fall risk, improving gait is one of the main goals during stroke rehabilitation [11].

To monitor progression and to support clinical decision-making, reliable gait assessments during rehabilitation are essential. In current practice, gait is assessed using several walking tests, e.g., the ‘6-minute Walking Test’ or ‘10-metre walk test’ [12,13]. These walking tests have a high clinical relevance since they reflect functional capacity, and because the outcomes are associated with fall risk [14,15]. However, gait features during the assessment, e.g., asymmetry and variability, are not routinely recorded. These features may provide additional insights in the individual walking dysfunction of people after stroke, leading to more accurate fall-prediction models, and presumably improving clinical decision making [7,16,17].

In recent years, several studies have demonstrated that portable devices, such as electromygraphy, insole foot pressure sensors, motion capture systems, and inertial measurement units (IMUS), can objectively measure gait and be used to compute gait features [18,19,20,21,22,23]. Of all of these devices, IMUs might be the most feasible for clinical use because measuring with IMUs requires no expensive equipment, and they are easy to operate. Numerous studies showed that gait features can be accurately determined using IMUs among the elderly, people with Parkinson’s disease, and people in the chronic phase after a stroke [24,25,26]. Moreover, IMU-derived gait features can be used to discriminate between fallers and non-fallers and different types of dementia [17,27]. In addition, several studies demonstrated that changes in gait features during rehabilitation, such as improved dorsi-flexion and symmetry, can be registered using IMUs [23,28]. This indicates that IMUs can be used to monitor gait recovery; hence, they function as a clinical evaluation tool. Despite the advantages of using IMUs to assess gait, regular IMU-based measurements have yet to be adopted by clinicians [21,29]. This is because IMU data needs to be collected, processed, and converted to clinically relevant information, requiring time, processing tools, and expertise. As a result, there is a sizable gap between research and clinical practice.

We aim to explore the potential of measuring gait using IMUs to closely monitor gait recovery in people after stroke. Therefore, as a first step, we determined whether gait features can reliably be obtained by clinicians in clinical rehabilitation. In the present study, we determined the test–retest reliability of IMU-based gait features during a two-minute walk test in people after stroke who were in clinical rehabilitation. Additionally, we examined whether these features are sufficiently reliable to be used to monitor individual progression. The key contributions of this study can be summarised as follows: firstly, we investigated if numerous gait features can reliably be measured using IMUs by clinicians in people after stroke; secondly, a stride-detection algorithm was developed to detect strides in the majority of people after stroke, including slow and asymmetric gaits; thirdly, a platform was created in which clinicians could upload measurement data that was then automatically stored, processed, and returned gait features; lastly, the protocol was designed in collaboration with clinicians to promote feasibility of assessing gait using IMUs in stroke rehabilitation.

## 2. Materials and Methods

### 2.1. Participants

Thirty-one people after stroke were recruited in two rehabilitation centres in the Netherlands. Participants signed a written informed consent prior to participating. All participants were diagnosed with stroke, according to the definition of the World Health Organisation [30], and had been hospitalised for four to fourteen days before admission. Eligible participants were above the age of 18, in the sub-acute or chronic phase after stroke, able to comprehend and sign the informed consent, and capable of understanding and performing simple tasks. The ability to understand and perform simple tasks was estimated by clinicians prior to inclusion. Participants were excluded if they were unable to walk at least 0.05 metres per second (≈seven metres in two minutes) or when they had experienced a recent (<4 weeks) thrombolysis or re-perfusion. The medical ethical review committee of Utrecht (METC number: 20-462/C) approved this study.

### 2.2. Protocol

Demographic- and stroke-specific characteristics were collected, including gender, age, time since stroke, type of stroke, and hemiparetic side. Additionally, outcomes of the following tests were obtained prior to the assessment: Berg Balance Scale, Trunk Control Test, Motricity Index, Modified ranking scale at admission, Barthel index at admission, and the functional ambulation classification with and without a walking aid [13,31,32,33,34,35]. In the assessment, participants walked at a self-selected speed for two minutes on a 14-m walking path with cones at both ends. Participants started on the left side of the starting cone and took right turns around the cones. The assessment was performed with a test–retest interval of 24 h. The use of a walking aid was optional. The assessments were administered by one trained physiotherapist per rehabilitation centre. A measurement was excluded if the subject stopped walking, was visibly distracted, or in case of an observable clonus. Participants were allowed to retry the assessment in case of a faulty measurement.

### 2.3. Equipment

Prior to the assessments, the gyroscope bias of each IMU was estimated using a fifteen-minute stationary measurement. Participants were equipped with three inertial measurement units (manufactured by Aemics b.v. Oldenzaal, The Netherlands). The inertial measurement units (IMUs) consisted of an accelerometer and gyroscope, and were measured with a sampling rate of 104 samples per second. The first IMU was placed on the lower back at a height of L5/S1. Its range was set to ±4 m/s^2^ and ±500°/s. The second and third IMUs were placed on top of the left and right foot, with the range set to ±8 m/s^2^ and ±500°/s. The IMUs were aligned with the anatomical axis during sensor placement. Elastic bands were used to hold the IMUs in place. Before and after the assessment, participants stood still for five seconds to enable an accurate assessment of the start and end of the assessment. The path and equipment are depicted in Figure A2 in Appendix D.

### 2.4. Data Processing

The data were imported and calculations were completed using Python (version 3.7.3). Firstly, the signal was down-sampled to 100 samples per second. Secondly, the gyroscope was corrected by subtracting the gyroscope bias. Thirdly, the first and last two seconds were excluded from further analysis to account for movement during transitional phases. Fourthly, the prior- and post-assessment stationary periods were estimated using a threshold based on the mean magnitude of the acceleration and gyroscope. Lastly, the length of the residual signals of all three IMUs were compared to the expected signal length. If the residual signal length deviated more than ten seconds from the expected signal length, the measurement was excluded from further analysis. These data-processing steps are visualised in Figure 1.

### 2.5. Stride Detection

To determine gait events in the left and right foot, a custom-made stride-detection algorithm was created, since existing stride-detection algorithms, such as continuous wavelet transform [36] and template matching [37,38], were inadequate for accurately detecting strides in very slow, poor, or asymmetric gait. To detect the strides, firstly, the average time per stride was estimated based on the dominant frequency found in the medio-lateral acceleration, using the Fast Fourier Transform [39]. Secondly, a peak-detection algorithm was used to identify foot-contact in the vertical acceleration (scipy.signal.find_peaks with a minimal distance of 0.75 * average time per stride and minimal height of mean vertical acceleration + standard deviation vertical acceleration). Subsequently, a false-negative peak detection was applied to minimise errors by comparing the sample difference between peaks to the expected between-peak difference (average time per stride). In the negative peak detection, the between-peak sample difference was evaluated; if the difference exceeded more than 1.5 times the expected difference, then the previously described peak detection was applied with 0.75 * the minimal peak height. After the false-negative peak detection, the stance phases were determined based on the stationary periods between foot-touches (peaks), where a stationary period was defined minimally as 0.2 consecutive seconds below a gyroscope- and acceleration-threshold (mean acceleration magnitude + standard deviation acceleration magnitude and mean gyroscope magnitude). In case no stance phase between foot-touches could be identified, a false-positive peak detection was applied to remove the lowest peak before determination of the stance phase. Swing phases were defined as the periods between stance phases. These processing steps are visualised in Figure 2. To compute spatial gait features, the accelerometer and gyroscope data were aligned with the vertical (VT) (upward: positive), medial–lateral (ML) (right: positive), and anterior–posterior (AP) axes (anterior: positive), and corrected for the effects of gravity using a sensorfusion algorithm [40]. Lastly, a zero-velocity update (ZUPT) was applied to determine spatial gait features [41].

For the determination of gait events in the low back, a similar algorithm was used as for the feet. First, the accelerometer and gyroscope data were aligned with the vertical (upward: positive), medio-lateral (right: positive) and anterior–posterior (forward: positive) axes, and corrected for the effects of gravity. Second, the anterior–posterior acceleration (AP) of the low back was integrated and filtered twice, with a second-order Butterworth bandpass filter between 0.25 and 15 Hz. Third, the first foot contact was detected using a peak-detection algorithm (scipy.signal.find_peaks with a minimal distance of 0.75 * average time per stride and minimal height of mean anterior–posterior acceleration + standard deviation anterior–posterior acceleration). Based on the mean outcome of the medio-lateral acceleration during the first step, the corresponding foot was determined [42]. Fourth, the time periods between foot contacts, found in the stride detection of the foot, were used as a template to detect all strides of the corresponding foot in the signal. Last, the first foot contact of the other foot was found using a peak-detection algorithm with a window between the first and second foot contact. The time periods between foot contacts, found in the step-detection algorithm of the foot, were used as a template to detect all strides of the corresponding foot in the signal.

### 2.6. Calculations

A total of 166 gait features were calculated, consisting of 56 spatio-temporal, 26 frequency, 63 complexity, and 14 symmetry features. A detailed description of all sway features is given in Table A3 and Table A4 in Appendix C. The spatio-temporal features were computed as the mean outcome per 10 strides. The paretic side and height were used for normalisation. If the paretic side was undefined (unknown or both sides affected), the left foot was used. The algorithm to process the data, detect strides and calculate gait features is available on GitHub: https://github.com/RichardFel/Reliability-of-Gait (accessed on 10 December 2021).

### 2.7. Statistics

The intraclass correlation coefficients and their 95% confidence interval for the between-day reliability were calculated using the intraclass correlation coefficient (ICC 2.1). An ICC of 0.5–0.75 was seen as moderate reliability, 0.75–0.9 as good, and 0.9 as excellent [43]. Additionally, the confidence interval (CI), standard error of measurement (SEM), and the minimal detectable change (MDC) were calculated. The MDC represents the threshold in which changes in score exceed measurement errors [44]. To determine the MDC independent of the unit of measurement, thus as a relative minimal detectable change, the MDC was divided by the standard deviation of the observed values of pooled test and retest measurements. This allows for comparison between features [45].

## 3. Results

### 3.1. Descriptives

Thirty-one people after stroke participated in the study. Participant characteristics are described in Table 1. Two participants were excluded, one because the required gait speed of 0.05 m per second was not met, and a second because of a clonus during the assessment.

### 3.2. Reliability

The ICC values of the test–retest measurements are visualised in Figure 3 and Figure 4. The mean, standard deviation, and ICC values are described in Table A1 in Appendix A. In total, 107 out of 166 gait features were measured with good–excellent reliability (ICC ≥ 0.75). These consisted of 50 out of 56 spatio-temporal, 8 out of 26 frequency, 36 out of 63 complexity, and 13 out of 14 asymmetry features. In total, 31 out of 46 gait features measured with the left foot IMU demonstrated good–excellent reliability. These consisted of 19 spatio-temporal and 12 complexity features. In total, 34 out of 46 gait features measured with the left foot IMU demonstrated good–excellent reliability. These consisted of 19 spatio-temporal, 3 frequency, and 12 complexity features. In total, 29 out of 54 gait features measured with the low back IMU demonstrated good–excellent reliability. These consisted of 12 spatio-temporal, 5 frequency, and 12 complexity features.

### 3.3. Clinical Monitoring

The relative minimal detectable change is visualised in the bottom panels of Figure 3 and Figure 4. The vast majority of relative minimal detectable change-values fell within approximately 0.5 and 1.5 standard deviations, indicating that a change of 0.5–1.5 standard deviations is minimally required to detect a change that exceeds the measurement error.

## 4. Discussion

We examined the test–retest reliability of various gait features using three inertial measurement units in people after stroke during clinical stroke rehabilitation. Additionally, the potential of these gait features to monitor progression was assessed. In summary, we found that the majority of the computed gait features were reliable and could potentially be used to monitor progression.

The gait features in four domains were computed, namely: spatio-temporal, frequency, complexity, and asymmetry. In line with the achieved results of [46], we found that the majority of the spatio-temporal features of the feet IMUs (stride time, stride distance, cadence) could be measured with high reliability. The achieved ICC values in our study were slightly higher, presumably because of the great between-subject differences. For example, some participants were able to walk only ten metres, where others walked more than one-hundred metres. Overall, the frequency features were less reliable than expected, with only 8 out of 26 features having a good–excellent reliability. This might be explained by the fact that the majority of frequency features, such as the dominant frequency width and density, are, in essence, measures of variability. Since only 2 min of walking were recorded, likely too few data points were collected to estimate these features with sufficient precision. Of all complexity features, only the autocorrelation and autocovariance demonstrated good–excellent reliability, whereas the Lyapunov exponent, sample, and approximate entropy demonstrated a poor–moderate reliability in both feet and the lower back. The low reliability of the Lyapunov exponent (LDE) in the lower back was particularly unexpected, since previous studies found this feature to be reliable [47,48]. The difference in reliability is presumably a result of the low number of strides included (25) in the computation of these features in our study. Increasing the number of strides in the analysis would have resulted in the exclusion of some slow and poor walkers; thus, these features seem unsuitable for measuring people after stroke in rehabilitation. The majority of the features regarding the swing-time and stance-time asymmetry demonstrated good–excellent reliability. These results are in line with the outcomes of the studies of Moore et al. [49] and Lewek and Randall [50].

The ICC values of the lower back features were considerably lower than the ICC values of the feet features. Most likely, this is caused by the low back sensor being subjected to significantly more noise than the feet sensors (e.g., clothing, trunk movement), and because the ground contact forces are damped before reaching the sensor, making the detection of gait events more difficult. This was especially true when measuring participants with severe gait impairments. Nevertheless, only a few of the computed gait features relied on the detection of gait events; thus, this did not affect the majority of the computed gait features.

To indicate the ability of gait features to register changes during clinical rehabilitation, the relative minimal detectable change was computed. Generally, a change of approximately 0.5–1.5 standard deviations is considered a difference that exceeds the measurement error and are thus related to a significant improvement or decline. Considering the fact that significant changes can be found in the walking ability, such as the walking speed, balance, and physical functioning during clinical rehabilitation, it is imaginable that some described gait features will be responsive to rehabilitation as well [51,52,53]. Nevertheless, A longitudinal study is imperative to determine if people after stroke are able to show significant improvements, reflected by these features during clinical rehabilitation.

Despite the good–excellent reliability of the majority of the features, this study has some limitations. First, only a relatively small number of people after stroke were included in the study. This may have caused the ICC values and MDC to lack precision. Nevertheless, if we evaluate the consistency of the ICC outcomes between features per sensor and between sensors, the outcomes appear to be robust. Secondly, participants that were not able to walk at least seven metres in two minutes (0.2 km per hour) could not be measured. Therefore, the conclusions may not generalise to all people after stroke in rehabilitation that are able to walk. Lastly, the custom-made step-detection algorithm, despite being extensively tested (Appendix B), was not validated for people after stroke in rehabilitation. However, the outcomes of the tests prior to this study and the consistency and reliability of the gait features calculated using the stride-detection algorithm provide an indication that stride detection is a valid method of detecting strides in people after stroke during rehabilitation.

Our ultimate objective is to develop an instrumented test that clinicians can use in daily practice to monitor individual progression during clinical stroke rehabilitation. For this reason, we tried to design a feasible protocol in which the majority of people after stroke can be measured, and by which sufficient information is collected, that is also easy to follow and time-efficient. Therefore, clinicians played an important role in the development of this protocol. Additionally, an online platform was created where clinicians could upload the IMU data, which was then automatically stored, processed, and converted into gait features. This platform allowed for direct feedback about the performance of the participant in comparison to other stroke participants. Presumably, the protocol and the platform improve the adaptation of clinicians to routinely measure gait using IMUs. Future work is in progress to determine if the computed gait features are sensitive to changes over time, and if these changes are of clinical importance. Eventually, the predictive value of IMU measurements during stroke rehabilitation on the levels of independence at discharge and fall risk will be examined.

## 5. Conclusions

This study examined the reliability of gait assessment using three inertial measurement units in people after stroke in clinical rehabilitation. In summary, we found that spatio-temporal, frequency, complexity, and asymmetry of gait features can reliably be measured during a two-minute walking test using a single measurement. Based on the relative minimal detectable change, it is likely that the proposed method can be used to monitor progression during stroke rehabilitation. 

## Figures and Tables

**Figure 1 sensors-22-00908-f001:**
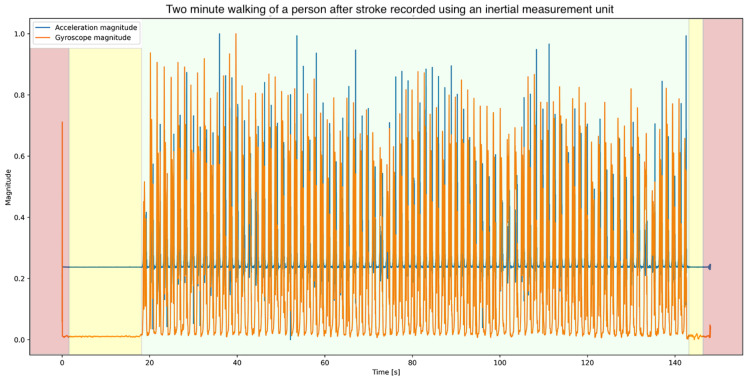
Visualisation of the data-processing steps prior to calculating the gait features. In the figure, the acceleration magnitude (blue) and gyroscope magnitude (orange) of a two-minute walking assessment is depicted. Firstly, the first and last two seconds (red planes) were excluded from analysis. The remaining signal was then used to calculate the threshold to determine the prior- and post-walking stationary periods (yellow planes). The residual signal was included in further analysis. For demonstrative purposes, the magnitude of acceleration and gyroscope were normalised using a min–max normalisation.

**Figure 2 sensors-22-00908-f002:**
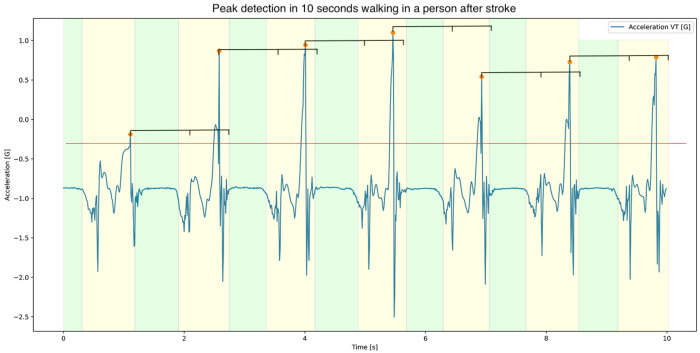
Visualisation of the custom-made stride-detection algorithm during 10 s of walking by a person after stroke. The vertical acceleration is depicted as a blue line. First, the strides were detected using a peak-detection algorithm; the threshold is depicted as the orange line and the search window is depicted as the black line above the peaks. The found peaks are marked with an orange circle. Then, a false-negative peak detection was applied, and the stance phases were identified as the stationary periods between peaks (green planes). Lastly, a false-positive peak detection was applied in case no stationary period between peaks could be identified. The signal that was not marked as part of the stance phase was considered to be the swing phase (yellow planes).

**Figure 3 sensors-22-00908-f003:**
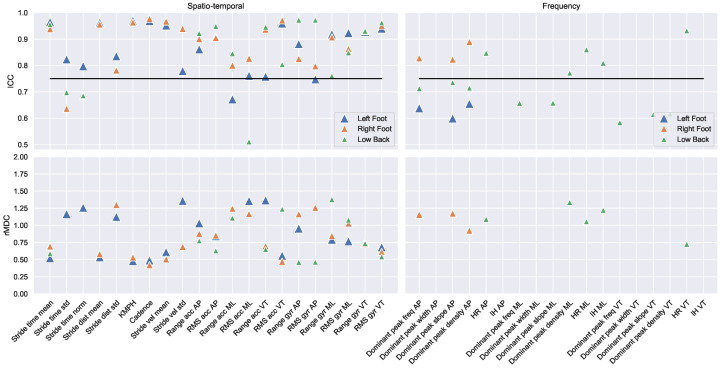
The ICC values (top panels) and rMDC values (bottom panels) of the spatio-temporal and frequency features for the left foot (blue triangle), the right foot (orange triangle), and the low back (green triangle). All outcomes above the horizontal black line were measured with good–excellent reliability. The exact outcomes of all gait features are provided in Table A1 in the Appendix A.

**Figure 4 sensors-22-00908-f004:**
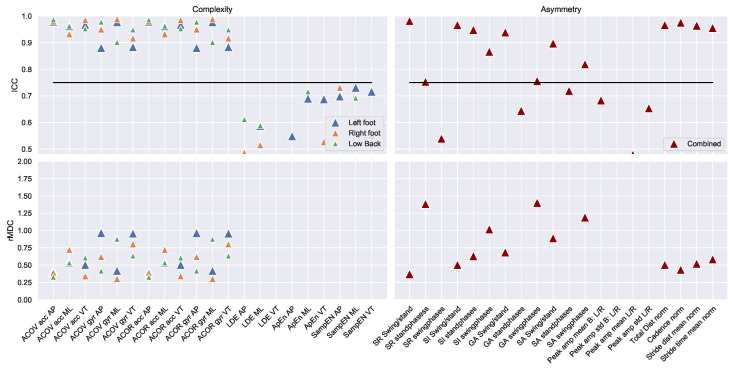
The ICC values (top panels) and rMDC values (bottom panels) of the complexity and asymmetry features for the left foot (blue triangle), the right foot (orange triangle), the low back (green triangle), and combined (red triangle). All outcomes above the horizontal black line were measured with good–excellent reliability. The exact outcomes of all gait features are provided in Table A1 in the Appendix A.

**Table 1 sensors-22-00908-t001:** Characteristics.

	Description	Outcome
Gender	Male/Female	15/15
Stroke type	Hemorrhagic/Ischimic	6/24
Hemiparetic Side	Left/Right/Both/Unknown	12/14/2/2
Walking aid	With/Without/Both	23/4/2
Age (years)	Mean (SD) (min, max)	69.2 (±10.3) [52, 85]
Time post stroke (weeks)	Mean (SD) (min, max)	10.4 ± 7.5 (3, 37)
Berg Balance Scale	Mean (SD) (min, max)	41 ± 11.7 (14, 56)
Motricity Index	Mean (SD) (min, max)	63.9 ± 32.3 (0, 100)
Trunk Control Test	Mean (SD) (min, max)	94.4 ± 16.2 (25, 100)
Barthel Index (at admission)	Mean (SD) (min, max)	10.3 ± 4.6 (1, 20)
Modified ranking scale (at admission)	Mean (SD) (min, max)	4.0 ± 0.7 (3, 5)
Functional ambulation classification	Mean (SD) (min, max)	2.1 ± 1.6 (0, 5)
Functional ambulation classification (walking aid)	Mean (SD) (min, max)	3.7 ± 0.8 (3, 5)

Abbreviations: SD = Standard deviation; Min = Minimum; Max = Maximum.

## Data Availability

Data will be made available on request in 2024.

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
