# Peer review of "Reliability of IMU-Based Gait Assessment in Clinical Stroke Rehabilitation"

_sensors, 2022, doi:10.3390/s22030908_

Round 1
Reviewer 1 Report
Large varieties of inertial sensors are currently available on the market, ranging from uniaxial accelerometers/gyroscopes to IMUs with 6 degrees of freedom. With the development of sensor technology and gait data analyzing techniques, gait analysis using wearable sensors has become a widespread and useful tool for both clinical practice and biomechanical research. Some detailed technical matters still need to be improved, such as the stability of sensor signals, the reliability of analytical algorithms for kinematics and kinetics in gait analysis.
Authors shall be clearly specified this part:
“To determine gait events in the left and right foot, a custom-made stride detection algorithm was applied, since the existing algorithms were inadequate of accurately detecting strides in very slow and poor walkers”
which algorithm are you referring to? Probably to the aemics motion tracker? Have any tests been conducted to affirm this? (study cited by the authors evaluated a different sensor)
This part is crucial to understand whether the study is an examination (test-retest reliability) of a custom made algorithm or an examination of a sensor commercially available and usable by Doctors.
Based on this I suggest changing the introduction, the title and adding more information to the statement “since the existing algorithms were inadequate of accurately detecting strides in very slow and poor walkers”.
Please, add photos on how the sensor was positioned (e.g. velcro) during the tests
Reviewer 2 Report
This study aimed to evaluate the reliability of IMU-Based Gait Assessment in Clinical Stroke Rehabilitation. I have the following major suggestions.
- Please add a paragraph about the contribution of this article in a bulleted form at the end part of the Introduction section.
- Which novelty do authors claim for this article? This IMU based gait assessment is widely studied approach in Clinical Stroke Rehabilitation.
- Authors claimed that a custom-made stride detection algorithm was applied. Authors should present claimed stride detection algorithm in detail with mathematical formulations.
- Authors should evaluate the reliability of stride detection using IMU with any standard stride calculation methods/equipment, as for example GaitRite.
- It's not clear, why authors utilized post-stroke participants? Are authors intending to find out anything special that is unique in the case of stroke patients? What would be different if authors would use healthy subjects?
- Authors should improve references mentioning the potential application of various gait studies in stroke rehabilitation https://doi.org/10.3390/s21165334, https://doi.org/10.1109/ICCE46568.2020.9043098. Background study needs to be improved.
- The results, discussion section need to be extended.
- Patient demographics showed that the average age (std.) was 69.2 (±10.3). Then is the age of participants in Figure A1 consistent (just curious) with patient demographics as described in Table 1?
- Captions of Tables and Figures need to be more detailed.
- Authors should discuss the strength and weaknesses of the proposed methods with other methods in the discussion section.
Round 2
Reviewer 1 Report
paper was improving according to suggestions
Reviewer 2 Report
Thanks for addressing the comments.